# The effect of gravity on hand spatio-temporal kinematic features during functional movements

Anna Bucchieri [1,2]*, Federico Tessari[3], Stefano Buccelli[1], Elena De Momi[2], Matteo Laffranchi[1], Lorenzo De Michieli[1]

1 Rehab Technologies Lab, Istituto Italiano di Tecnologia, Genoa, Italy, 2 Department of Electronics, Information and Bioengineering, Politecnico di Milano, Milan, Italy, 3 Newman Laboratory for Biomechanics and Human Rehabilitation, Massachusetts Institute of Technology, Cambridge, Massachusetts, United States of America

* anna.bucchieri@inria.fr

**Data Availability Statement:** The dataset has been officially published on IIT Dataverse platform. Here is the information: Bucchieri, Anna, 2024, "Kinematic dataset for an unconstrained pick and

## Abstract

Understanding the impact of gravity on daily upper-limb movements is crucial for comprehending upper-limb impairments. This study investigates the relationship between gravitational force and upper-limb mobility by analyzing hand trajectories from 24 healthy subjects performing nine pick-and-place tasks, captured using a motion capture system. The results reveal significant differences in motor behavior in terms of planning, smoothness, efficiency, and accuracy when movements are performed against or with gravity. Analysis showed that upward movements ($g^-$) resembled transversal ones ($g^0$) but differed significantly from downward movements ($g^+$). Corrective movements in $g^+$ began later than in $g^-$ and $g^0$, indicating different motor planning models. Velocity profiles highlighted smoother movements in $g^-$ and $g^0$ compared to $g^+$. Smoothness was lower in $g^+$, indicating less coordinated movements. Efficiency showed significant variability with no specific trends due to subjective task duration among subjects. This study highlights the importance of considering gravitational effects when evaluating upper-limb movements, especially for individuals with neurological impairments. Planning metrics, including Percent Time to Peak Velocity and Percent Time to Peak Standard Deviation, showed significant differences between $g^-$ and $g^0$ compared to $g^+$, supporting Fitts' law on the trade-off between speed and accuracy. Two novel indications were also introduced: the Target Position Error and the Minimum Required Tunnel. These new indicators provided insights into hand-eye coordination and movement variability. The findings suggest that motor planning, smoothness, and efficiency are significantly influenced by gravity, emphasizing the need for differentiated approaches in assessing and rehabilitating upper-limb impairments. Future research should explore these metrics in impaired populations to develop targeted rehabilitation strategies.

place task", https://doi.org/10.48557/KXUZIW, IIT Dataverse, V1.

**Funding:** This work was supported by Istituto Nazionale per l'Assicurazione contro gli Infortuni sul Lavoro, under grant agreement "PR19-RR-P2-RoboGYM".

**Competing interests:** The authors have declared that no competing interests exist.

## Introduction

Able-bodied humans are capable of performing a wide array of complex and challenging motor activities throughout their lives. Either art, sport, or labour-related, all actions are inevitably influenced by one constant environmental parameter: gravity. Indeed, starting from their first day of life, humans experience the need of controlling their own body while immersed in the gravitation field and interacting with objects of different inertial properties. Thanks to adaptation processes, adults are then capable of performing fundamental activities in their daily lives to maintain personal well-being and independence.

Physiologically, throughout human development, the brain is characterized by a process called neuroplasticity, where neural connections adapt in response to environmental changes. This allows the phenomenon of learning involving the acquisition of new motor planning and execution abilities [1]. The brain creates a cognitive representation of the body and its interactions with the external surroundings. Such framework, called internal model, allows to predict how the body will react to actions, movements, and sensory inputs [2, 3]. Particularly, an internal kinematic model translates information from the task space (i.e. hand trajectory for the upper-limb) to the joint space, meanwhile an internal dynamic model computes the joint torques needed to perform the given activity [2, 4]. Nonetheless, the motor planning process results in both kinematic and dynamic constraints on the execution of the movement [5, 6].

The central nervous system considers the mechanical effects of gravity prior to a specific task execution [7], thanks to the activation of several somatosensory channels in charge of the "graviception" [3, 8, 9]. This is particularly noticeable for vertical tasks where path execution differs between upward and downward movements [10] and was proven in microgravity studies (i.e. Zero-G). The motor behaviour observations of subjects in parabolic flights revealed, for the first few attempts, the same direction-dependent changes in the upper-limb kinematics with respect to earth gravitational conditions (i.e. One-G) [4, 11]. Then, when adaptation to the new environment occurred, the hand paths slowly tended to become straighter and similar between upward and downward movements [4, 7, 12], meanwhile postural synergies were greatly affected by microgravity leading to loss of body-coordination [13]. Once back to One-G condition, motor plans tend to adapt back to the pre-flight conditions. As anticipated, the execution of movement depends also on the proprioception and visuo-perception of the body in the environment [3, 8, 9]. These systems enrich the musculoskeletal dynamics of the internal model, especially when interacting with objects, allowing the central nervous system to counter the influence of the load on both the acceleration and the gravitational terms of the upper-limb [11, 14].

Nonetheless, such ability is lost in patients who lack proprioception, resulting in abnormal reaching movements and hand trajectories [6, 15–18]. Up to 69% of post-stroke survivors—leading cause of disability worldwide [19]—experience a loss in proprioception [20], limiting motor learning abilities [21] and independence in activities of daily living [22–26].

Everyday tasks are mostly performed in the frontal plane and require large glenohumeral elevations [27]. The earlier discussions emphasize that upward and downward movements are the result of different neurological schemes within the central nervous system. Consequently, the rehabilitative intervention should also align with each respective direction. A proper quantification of hand postural patterns (defined as the characteristic way in which a body part moves) when moving against or propelled by gravity could offer a significant insight on activity of daily living recovery. To better understand the effects of neurological disorders on motor control, it is fundamental to first study healthy behaviors. The observation of 2D point-to-point tasks allowed motor control researchers to identify key features of

healthy hand patterns [28–31]. Krebs and colleagues [32] provided evidence that complex movements can be deconvolved into movement primitives—each characterized by bell-shaped speed profile—mildly impacted by neurological disorders. Throughout physical rehabilitation, submovements tend to blend together leading to smoother hand patterns [32, 33], thereby making them a fitting tool for motor learning evaluations [34]. Indeed, healthy kinematic features—calculated from such patterns—can be exploited as reference metrics to assess the level of impairment of patients [33]. Integrating clinically valid scales (i.e. Fugl-Meyer, Action Research Arm Test) with performance metrics could widen up the field of view on the patient's impairment state, offering detailed insights on underlying aspects of a given task [35, 36], as well as differentiate between genuine motor recovery and compensation [35, 37–40].

Systems such as the MIT-Manus [41] allow to standardize planar movements imposing repeatable constraints across trials and subjects [28, 29, 42–44]. On the contrary, the analysis of 3D movements is more complex and articulated, even exploiting upper-limb exoskeletons [45, 46]. The assessment of everyday activity recovery is still an open challenge due to the lack of proper scales and standardized evaluation strategies [35, 47–51]. Such absence can be attributed to the introduction of novel features without the prior validation of pre-existing ones [36]. Indeed, kinematic metrics tested in literature are diverse and not often superimposed. The most commonly used performance scales are related to smoothness, efficiency, movement planning, and accuracy [52, 53] but little attention was placed upon the effect of movement direction changes on quantitative indices and their clinical relevance [54]. Few available examples report the analysis on post-stroke and healthy subjects for 2D [52, 55] and 3D [53, 56] tasks, where a direction-dependency on the execution of reaching movements and a correlation between kinematic metrics and Fugl-Meyer were found [51, 57].

Overall, the state-of-the-art is focused on free-reaching and reach-to-grasp tasks, where the effects of forward and backward movements on clinimetrics are studied. However, self-care and routine activities not only require large movements in the coronal plane, but also very often physical interaction with objects. This aspect should not be overlooked in occupational therapy, where a proper evaluation of functional recovery plays a pivotal role in customizing the rehabilitation process and ensuring the restoration of personal independence.

The present study specifically tackles this problem. Starting from a standardized pick-and-place task—selected as a simplified version of a common everyday activity—an experimental study was conducted to observe healthy hand trajectories while performing 9 pick-and-place movements—6 in the frontal plane (i.e. 3 upward and 3 downward) and 3 in the transverse plane.

Based on the previous knowledge of gravitational influence in performing a task, a special effort was made to characterize the Cartesian behaviour of participants' hand in the context of existing motor-control literature, with the aim to quantify how hand patterns change when moving against/propelled by gravity and interacting with physical objects. Particularly, five metrics were selected according to their frequent use in research studies and correlation with clinical scales, and two additional novel features were introduced to better characterize the proposed task.

The quantification of kinematic indices is thoroughly discussed with respect to the gravity-related knowledge on motor executions. The existence of a central tendency in the analyzed features is examined to understand the possibility to exploit the proposed task as a platform for the evaluation of activities of daily living upper-limb recovery. The results identify a set of kinematic features that could be potentially used in upper-limb motor assessment.

## Materials and methods

### Ethics statement

All experiments were conducted in accordance with the Declaration of Helsinki. Participants signed a written informed consent and followed the ethical protocol "IIT REHAB HT01 (363/ 2022)—DB id 12494" approved by the Ethical Committee of Liguria Region in Genoa, Italy. Experimental sessions started on 2nd December 2022 and finished on 7th December 2022.

### Participants

In this study 24 healthy right-handed subjects have been recruited (13 females, 12 males) between 25 and 45 years-old. People from the 23rd female to 99th male height percentiles were selected in order to investigate the performance of movements in a highly variable subjects' population.

### Experimental set-up and pre-processing

A Motion Capture (MoCap) analysis has been performed to record subjects' upper-limb movements in space with a sample frequency of 100Hz. Particularly, the Vicon Nexus 2.12.1 was used. Subjects were equipped with 12 infra red reflective markers as shown in (Fig 1A and 1B). The MoCap software exploits the 12 markers to create a user-specific biomechanical model thanks which missed samples during the acquisitions can be identified and reconstructed. MoCap cameras were calibrated following Nexus Vicon guidelines. The calibration was performed for each subject to ensure accuracy in kinematic data acquisition of the reflective markers. The global reference frame was set on the floor in correspondence to the bottom left corner of the custom-made library (Fig 2).

The task consists in a matrix of 9 pick-and-place movements (M1-M9, Fig 1C) performed on a customized library (Desk is a surface of 40x80 cm, shelf of 20x80 cm). To standardize the task, the shelf was aligned with each subject's shoulder height, meanwhile the table was placed 40 cm lower regardless from subjects' height percentile. The distance between the targets, recreated on paper, was fixed for all subjects. Fig 1C shows a representation of the counterclockwise order in which each box was picked from a starting position and placed on a target. M1-M3 are performed against gravitational force thus they will be referred as $g^-$, M4-M6 propelled by gravity and referred as $g^+$, and M7-M9, performed on the transversal plane, referred

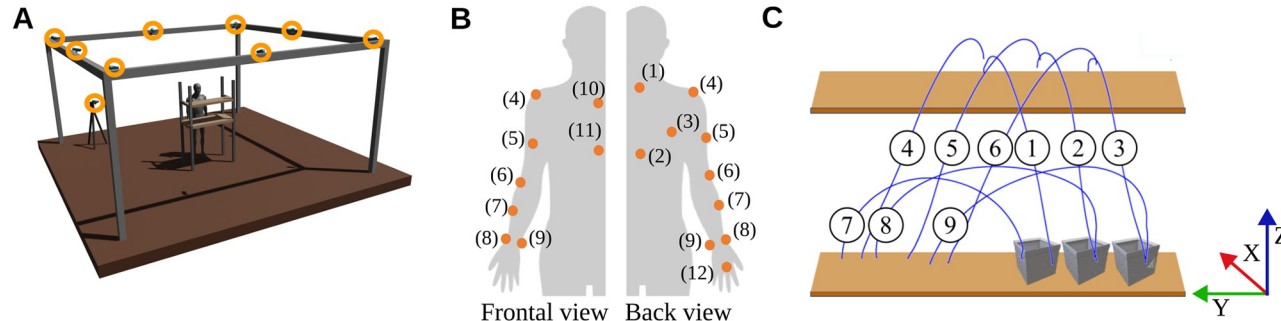

**Fig 1. Experimental set-up.** (A) Environment: 9 Nexus Vicon infrared cameras (orange circles) and a custom-made library made of a table and a shelf. (B) Positioning of IR-reflective markers on subjects (1)C7, (2)T10, (3)Scapula, (4)Shoulder, (5)Arm, (6)Elbow, (7)Forearm, (8)Outer Wrist, (9)Inner Wrist, (10) Collar bone, (11)Sternum, (12)Knuckle (C) Proposed pick-and-place task: 3 movements from the desk to the shelf (M1-M3), 3 from shelf to desk (M4-M6), and 3 from desk to desk (M7-M9).

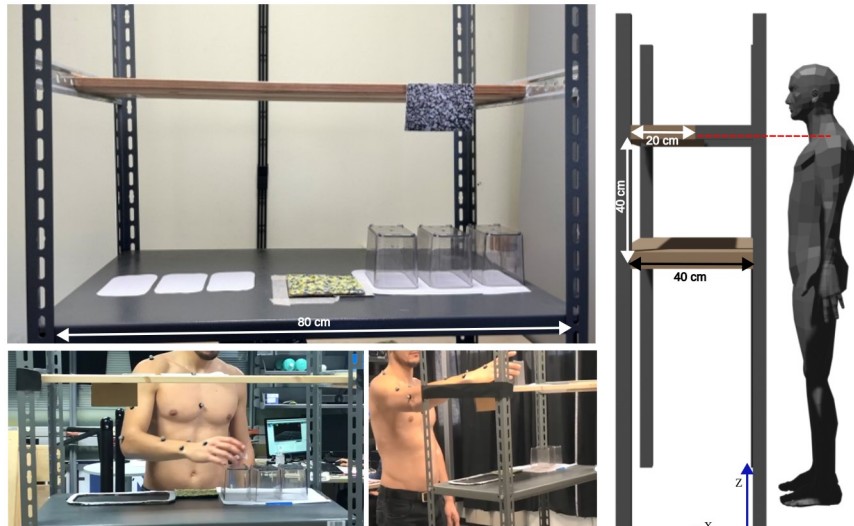

**Fig 2. Physical set-up and a participant performing the proposed pick-and-place task.**

as $g^0$ (Table 1). The main amplitude of movements was 40 cm for g- and g+ movement. The desk width was 80 cm but g0 movements were also constrained to have an amplitude of 40 cm (Fig 2).

Subjects were asked to move the boxes from target to target following a counter-clockwise order. Each movement was recorded separately from the others. Indeed, participants were asked to keep their right upper-arm along the body, pick the object, place it on shelf, and go back to the resting position. Recordings were started and ended at each resting phase. The acquired data were firstly processed on the Vicon Nexus software. Markers trajectories were filtered with a 4th-order Butterworth low-pass filter, cut-off frequency of 6Hz.

As the designated task comprised three distinct movement phases—initial reach-to-grasp, pick-and-place, and return to the resting position—standard isolation techniques based on velocity profiles could not be applied to identify the interested movement [58–61]. Therefore, the trajectory outlines in Cartesian space were utilized. Given that the library surfaces maintained a fixed relative distance along the Z-axis, the point of contact with the desk marked the initiation of the picking phase. Conversely, for upward movements, the moment of contact with a shelf positioned 40 cm above the target shelf indicated the commencement of the placing phase; conversely, for downward movements. Likewise, for lateral movements, the return to a height of 40 cm signaled the initiation of the placing phase. Subsequently, each trajectory was segmented at these critical time points. These processes were executed in MATLAB R2021b.

**Table 1. Labels for the proposed 9 pick-and-place movements.** Movements performed in the same direction where clustered together and labeled under one name.

| Gravity Condition | Movements | Label |
|---|---|---|
| Against | M1, M2, M3 | $g^-$ |
| Propelled | M4, M5, M6 | $g^+$ |
| Neutral | M7, M8, M9 | $g^0$ |

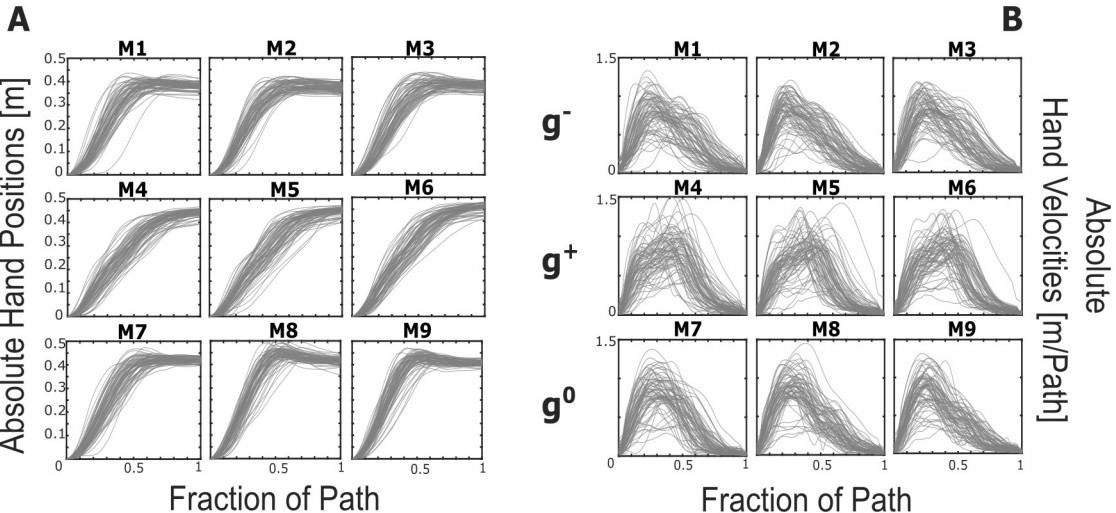

**Fig 3. Representations of absolute hand positions (A) and velocities (B) across each subject, repetition, and movement.** Each curve was normalized over their own duration thus velocity is expressed as m/path. M1-M3 can be grouped as g-, M4-M6 as g+, and M7-M9 as g0.

## Data processing

Extracted MoCap data were processed using MATLAB R2021b. Trajectories of marker 12 (Fig 1B) were related to the hand movements. Absolute hand positions were calculated as

$$P_m^{s,r} = \sqrt{X^2 + Y^2 + Z^2} \tag{1}$$

Where m is the movement (M1-M9), s is the subject (S1-S24), and r is the repetition (R1-R3). Velocities were derived from the hand positions: hand positions were firstly filtered with a moving average of 10 Hz to reduce experimental noise, and then differentiated with a central difference derivative. Absolute hand velocities were calculated as

$$V_m^{s,r} = \sqrt{V_X^2 + V_Y^2 + V_Z^2} \tag{2}$$

Absolute hand paths and velocities were normalize over their own timing (Fig 3).

MATLAB Curve Fitting tool was used to check the polynomial better fitting hand positions. Root-Mean-Squear Error was calculated to compare experimental and fitting curves

$$RMSE = \sqrt{\frac{1}{n}\sum_{i=1}^{n}(y_i - \hat{y}_i)^2} \tag{3}$$

$n$ represents the number of data points. $y_i$ represents the actual (observed) value of the dependent variable for the $i$-th data point. $\hat{y}_i$ represents the predicted (estimated) value of the dependent variable for the $i$-th data point.

## Metrics

In this study 7 kinematic metrics were selected related both to hand displacements and velocities. Following examples in literature [36, 53, 57], the features can be expressed in terms of smoothness, planning, efficiency, and accuracy. For each category different metrics were either selected from literature or introduced as novel indicators and Table 2 groups together the name, acronym, and reference for each performance feature.

**Table 2. Seven kinematic features were selected according to 4 different performance evaluations: Smoothness, planning, efficiency, and accuracy.** Abbreviations are reported, as well as literature references, if present.

| Performance | Metric | Abbreviation | Reference |
|---|---|---|---|
| Smoothness | Spectral Arc Length | SPARC | [36, 54, 56, 57, 62] |
| Smoothness | Number of peaks in velocity | NVP | [33, 34, 36, 52–54, 57] |
| Planning | Percent Time to Peak Velocity | PTPV | [36, 51, 53] |
| Planning | Percent Time to Peak Standard Deviation | PTPSD | [64] |
| Efficiency | Movement Time | MT | [36, 53, 57, 64] |
| Accuracy | Target Position Error | TPE | Novel |
| Accuracy | Minimum Required Tunnel | MRT | Novel |

**Smoothness.**

- Spectral Arc Length (SPARC): Described as the Arch Length of the frequency spectrum obtained by applying a Fourier Transform of the velocity profile for a movement, within a dynamically defined frequency range. This metric is negated so that more negative numbers represent less smooth data. Several studies employed SPARC metric to assess upper-limb impairments [36, 62]. Particularly Bayle et al. [56] performed a comparison between 4 different smoothness metric (i.e. SPARC, number of zero-crossings in the acceleration profile (N0C), log dimensionless jerk (LDLJ), and normalized average rectified jerk (NARJ)) during a 3D reaching point-to-point task, and defined SPARC as the most significant one. The systematic review of Mohamed and colleagues [54] reached similar conclusions out of 32 selected metrics. Hajihosseinali et al. [57] performed a reliability test (Intraclass Correlation Coefficient—ICC) across 24 kinematic metrics and 2 clinimetrics (i.e. FM and ARAT) and found SPARC passed the test, as well as the study of Saes et al. [63] for 2D horizontal reaching movements.

$$SPARC = -\int_{f_{min}}^{f_{max}} \left| \mathcal{F}\left\{ \frac{dv(t)}{dt} \right\} \right|^2 df \qquad (4)$$

- Number of Velocity Peaks (NVP): this metric was also widely used in literature as a representation of the number of submovements needed to perform an action. Studies of [33, 34] showed that post-stroke survivors hand patterns reflect several peaks in the speed curve when performing horizontal point-to-point reaching tasks. Following a rehabilitative period, submovements tend to grow larger and blend in number, thus the speed profile converges toward a bell-shape trend typical in the healthy behaviour. As a result, the authors suggest NVP to be a quantitative marker of neurological recovery. This metric does not present a unique terminology across the state-of-the-art (i.e. Num Peaks or $n_{peaks}$ [53], No. of velocity peaks [36], N_submov [33], Peaks [54], nPS [52], nPK [55], and NVP [57]). In this manuscript NVP was employed. As anticipated, experimental hand positions underwent filtering prior to derivation and velocity profile calculation. To mitigate the impact of noise during peak detection, specific criteria were applied based on empirical considerations. Particularly, peaks were considered local maxima if their height $h_i$ satisfied the condition:

$$h_i \geq \frac{1}{2} \cdot h_{\max}, \qquad (5)$$

where $h_{\max}$ is the maximum peak height. Additionally, peaks were required to be separated

from any preceding peak by a distance ($d_i$) of at least 10% of the total path length ($L$):

$$d_i \geq 0.1 \times L. \tag{6}$$

**Planning.**

- Percent Time to Peak Velocity (PTPV): The terminology of this metric is not unique (i.e. Control Strategy [51], Time to Peak Velocity [36]) thus the one introduced by [53] was adopted in this manuscript. PTPV is defined as the fraction of path at which maximum speed ($V_{\max}$) is detected.

$$\text{PTPV} = \frac{\text{Path length at } V_{\max}}{\text{Total path length}} \cdot 100 \tag{7}$$

- Percent Time to Peak Standard Deviation (PTPSD): Defined as the fraction of path at which hand displacements across subjects and trials show maximum standard deviation ($\sigma_{\max}$), averaged over x, y, and z dimensions [64].

$$\text{PTPSD} = \frac{\text{Path length at } \sigma_{\max}}{\text{Total path length}} \cdot 100 \tag{8}$$

**Efficiency.**

- Movement Time (MT): This metric has widely been linked to movement efficiency. MT is defined as the duration of the movement ([36, 53, 64, 65], DUR [57]) from when the object is picked ($t_{start}$) to when it is placed on target $t_{end}$.

$$\text{MT} = t_{end} - t_{start} \tag{9}$$

**Accuracy.** Accuracy metrics employed in the state-of-the-art are mostly related to point-to-point reaching tasks either in 2D or 3D and assume straight paths (i.e. Initial Detection Error, End point error, straight line deviation [36, 53, 62]). The same assumption can't be advanced for the pick-and-place task proposed since the hand trajectories are visibly curved and the placement of an object on target is required. As a result, two novel metrics are introduced to depict the variability across subjects and trials while performing the movement (Minimum Required Tunnel), and placing the box (Target Position Error).

- Target Position Error (TPE): Defined as the radius of the smallest sphere containing up to 95% of the final hand positions for each set of movements. Such marker could be useful in determining the level of impairment of neurological patients in accurately positioning an object on target (i.e. if the end-point falls inside or outside the spheres across repetitions) and check whether throughout the therapeutic journey the patient-specific sphere converges into healthy dimensions.

- Minimum Required Tunnel (MRT): This metric actually holds two possible measurements related to 1 standard deviation (accounting for 67% of intra-subjects variability) and 3 standard deviations (accounting for 99.7% of intra-subjects variability):

  - Area: given the mean three-dimensional standard deviation at each time-normalized hand path across subjects and trials ($\sigma_{mean}$), MRT is defined as the area under the trend of 3

times the standard deviation ($3\sigma_{\mathrm{mean}}$) minus 1 standard deviation ($\sigma_{\mathrm{mean}}$):

$$\mathrm{MRT} = \int_{t_{start}}^{t_{end}} (3\sigma_{\mathrm{mean}}(t) - \sigma_{\mathrm{mean}}(t))\, dt \tag{10}$$

- 3D volume: In a previous work [66] it was introduced an algorithm to create reference tunnels of variable radius centered on the average of hand paths across subjects and trials. The variable radius is given by the standard deviation and can be multiplied to any factor to scale its volume. Such metric can serve as a marker of how many standard deviations are needed to include upper-limb movements of subjects suffering an impairment, and check whether the minimum required tunnel decreases through the rehabilitative process.

## Statistical analysis

Statistical analysis was performed with MATLAB R2021b.

Multigroup comparison tests were employed to check differences among means, regardless of whether global tests were significant, due to the interest in examining comparisons between specific groups. In statistics, when testing interaction effects, omnibus F tests may not fully explain the source of the interaction, especially if factors involved have more than two levels. Instead, contrasts are used to examine specific differences between these factors. Employing a priori interaction contrasts to test hypotheses derived from theory offers a more focused approach compared to omnibus tests, potentially yielding deeper insights into patterns of data [67, 68].

Normality of the dataset was assessed employing the Jarque-Bera test.

Given the repetitive nature of the experimental protocol, the Friedman non-parametric test was employed, while repeated measures ANOVA was applied to normally distributed datasets. Considering the acquired dataset, some metrics showed a quasi-normal distributed trend. Even though repeated measures ANOVA test relies on normal distribution assumptions, F statistic can be considered robust, and samples can deviate from normality, if they exhibit symmetry or similar shapes, particularly when sizes are equal and sufficiently large [69].

Pairwise post-hoc comparisons were conducted using Mann-Whitney (non-normally distributed) and t-tests (normally distributed) to analyze various measurements within the same set. Considering that each condition involved three comparisons, a Bonferroni correction was implemented [70] and the significance level was reduced to $\alpha = 0.05/3 = 0.017$.

## Results

Participants were instructed to perform nine distinct movements (M1-M9) during a Motion Capture (MoCap) analysis (acquisition rate of 100Hz, Fig 1A and 1B). The task selected was the pick-and-place one: each subject was required to grab three boxes (10x8x9.5 cm, 0.2 kg) consecutively and move them onto different locations on a library, made of a table and a shelf. Each sequence underwent three repetitions and no limitations on timing and paths were imposed. Indeed, subjects were asked to perform the movements in a relaxed and natural way. No Particular information was give about the task goals and aims to keep them as naive as possible.

It's worth highlighting that, since boxes featured a rectangular cross-section, corresponding targets placed on the library were recreated on paper to match this shape. As a consequence, the orientation at which objects had to be placed onto marks was constrained, resulting in

limitations in wrist displacements. Such design choice was purposely adopted to enhance glenohumeral and elbow joint movements. Average recorded standard deviation of wrist displacements across subjects, repetitions, and movements is 1.2˚ for flex/extensions, 8.4˚ for prono/supination, and 4.4˚ for ulnar-deviations respectively.

For each movement subjects were required to keep their arm on a resting position alongside the trunk, reach and grasp the box, move it onto target, and get back to the resting position. The primary objective of this study is to exclusively observe Cartesian trajectories of the hand (Fig 1B, marker 12 placed on the knuckle of the right middle finger) while performing the pick-and-place movement (further details on the experimental set-up and data processing in Section Materials and Methods).

Fig 3A reports the evolution of absolute hand displacements for each subject, repetition, and movement, normalized over their own movement time. The observed curves can be modeled as 6th order polynomials with an average root-mean-square-error of 3.3 ± 1.9 mm. Independently of the movement direction, the curves show a similar trend made of two distinct phases: an initial part related to hand motion, and a second plateau stage where hand displacements remain rather constant with respect to path fraction. This second phase could be an indicator of a corrective strategy for the placement of the object on target. Similarly, absolute hand velocity profiles (Fig 3B) resemble asymmetrical bell-shaped trends for all movements.

Clustering together movements performed in the same direction, $g^-$ and $g^0$ present a similar trend. Instead, $g^+$ shows a less remarked correction phase. $g^{-,+}$ are vertical movements, thus an empirical threshold of 5% of the maximum peak velocity [49] was applied to the speed profile in the z-axis, meanwhile for $g^0$ to the speed profile in the y-axis was considered, as the movement is mainly performed in the Transversal plane. On average $g^-$ and $g^0$ first reached velocity lower than the threshold at 57% and 62% of the path respectively, meanwhile $g^+$ at 87%.

Fig 3B presents the absolute hand velocity profiles for each movement. Also in this case the trend is comparable across subjects and repetitions, as well as the overall behavior of $g^-$ and $g^0$ with respect to $g^+$. Indeed, M1-M3 and M7-M9 are characterized by an initial steeper slope than the descending one. M4-M6, instead, seem qualitatively mirrored, suggesting that, when present, a local maxima is reached before the maximum peak and vice versa for $g^{-,0}$ (Fig 4A, in red maximum peaks and in blue subsequent ones). Out of the total number of curves (216) for each direction-dependent movement, the majority of them display a singular peak (67% for $g^-$, 53% for $g^+$, and 66% for $g^0$). It is noteworthy that while both sets $g^-$ and $g^0$ manifest a similar distribution of maximum peak counts across repetitions and subjects (denoted as Number of Velocity Peaks—NVP Table 2), set $g^+$ deviates slightly in this regard.

Within the latter, the incidence of curves featuring one or two peaks is more evenly balanced (114 versus 93, Fig 4B). Friedman test calculated over the three gravity conditions $g^-$, $g^+$, and $g^0$ resulted in a statistically significant difference (p < 0.001*, $\chi^2(23) = 154$, $\eta_p^2 = 0.27$). However, pairwise Mann-Whitney tests performed over $g^- - g^+$ and $g^+ - g^0$ resulted in statistically significant differences, presenting p = 0.006 and p = 0.011 respectively, meanwhile comparison between $g^- - g^0$ did not yield any statistically significant result as p = 0.86 (Significance level corrected to 0.017 following Bonferroni approach [70] Table 4).

From the presented outlines, 6 additional kinematic metrics were calculated (Table 2). Particularly, the smoothness of the movement was evaluated through the Spectral Arc Length (SPARC), the planning of the movement was related to the Percent Time to Peak Velocity (PTPV), and Percent Time to Peak Standard Deviation (PTPSD), and the efficiency to Movement Time (MT). Given the complexity of the task within the 3D space, adapting accuracy metrics usually applied in reaching tasks proved challenging and less applicable in the current context (i.e. trajectory error, end point error [36], straight line deviation [62]). As a result, two

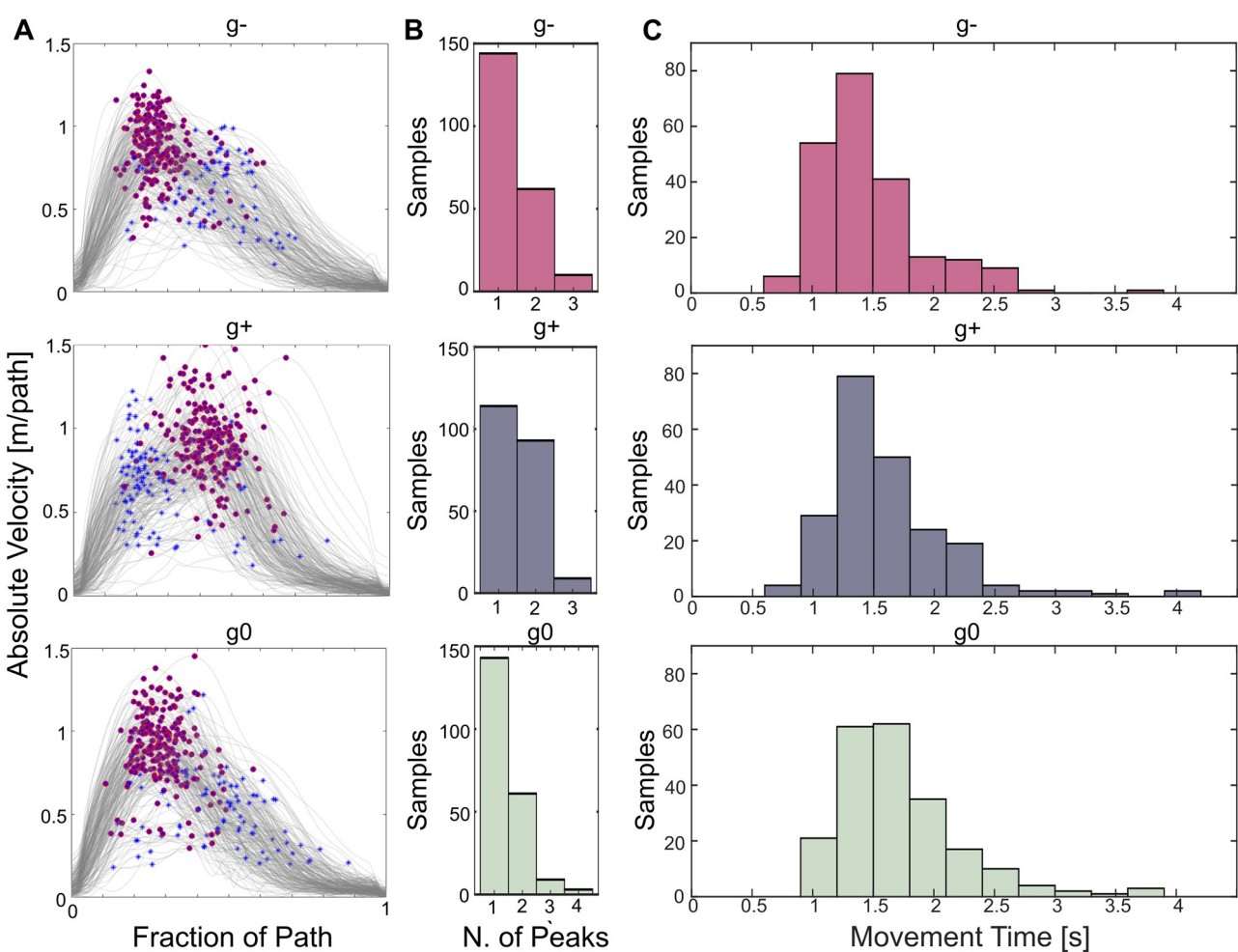

**Fig 4. Representation of velocity peaks, number of peaks in the velocity profile and Movement Time distribution for each gravity condition.** (A) Each row represents the absolute hand velocities for either $g^-$, $g^+$, and $g^0$ sets of movements. The maximum peak is marked in red meanwhile all the local maxima are marked as blue stars. (B) Distribution of number of velocity profiles presenting 1, 2, 3, or 4 local maxima. (C) Movement Time (MT) distribution for each gravity condition. The arrangement is positively skewed for $g^-$ (1.43), $g^+$ (1.77), and $g^0$ (1.44).

novel indices are presented as Target Position Error (TPE) and Minimum Required Tunnel (MRT).

Statistical analyses related to Percent Time to Peak Velocity, Percent Time to Peak Standard Deviation, Spectral Arc Length, and Movement Time were conducted to assess, if present, any direction-dependency with respect to gravitational force. Particularly, Movement Time, as for Number of Velocity Peaks (previously discussed), does not exhibit a normal distribution therefore Friedman non-parametric test was employed to determine any significant differences among the repeated measures of the metric (significance level 0.05). For the PTPV, PTPSD, and SPARC metrics—since they all presented unimodal and predominantly symmetric distributions—repeated measure ANOVA was employed to investigate the effect of direction (significance level 0.05).

Additionally, post-hoc tests were conducted to quantify the difference between each condition. Since 3 comparisons are performed for each metric and hypothesis, the significance level was adjusted to 0.017 following the Bonferroni correction to avoid risks of Type I error when

**Table 3. Kinematic metrics calculated from 24 healthy subjects while performing 9 pick-and-places movements.**

| Condition | NVP(IQR) | MT(IQR)[s] | PTPV ($\sigma$) | SPARC ($\sigma$) | PTPSD ($\sigma$) | TPE [cm] | MRT [cm] |
|---|---|---|---|---|---|---|---|
| $g^-$ | 1(1) | 1.38(0.43) | 26.9(8) | -1.54(0.057) | 33(11.4) | 3.33 | 3.84 |
| $g^+$ | 1(1) | 1.49(0.53) | 42.3(8.7) | -1.6(0.068) | 44.5(15.9) | 3.3 | 4.25 |
| $g^0$ | 1(1) | 1.64(0.53) | 26.75(6.6) | -1.54(0.057) | 34.2(9.1) | 2.3 | 4.69 |

NVP: Number of Velocity Peaks, MT: Movement Duration, PTPV = Peak Time to Peak Velocity, SPARC = Spectral Arc length, PTPSD = Peak Time to Peak Standard Deviation, TPE = Target Position Error.

making multiple statistical tests [70]. For more in-depth information regarding the metrics and statistical tests used, please consult the Materials and Methods section.

Metrics TPE and MRT were computed across all trajectories per type of movement, thus for each gravity condition only three evaluations are available. As no statistical tests or standard deviation could be computed, TPE and MRT are mostly regarded as indicators of task-specific global accuracy in healthy subjects.

Considering each movement (i.e. M1-M9), both absolute hand positions and velocities are comparable across subjects and trials within the same gravity condition.

The proposed kinematic features (Table 2) are not influenced by the start and target position of the 3 objects (M1-M9) as statistical analysis on $g^-$, $g^+$, and $g^0$ did not show a dependency on location of the load (p > 0.05) for all metrics but MT. Following these conclusions, data relative to movements of the same gravity condition were pooled together (M1-M3 for $g^-$, M4-M6 for $g^+$, and M7-M9 for $g^0$, Table 1). Interested readers can refer to the Supplementary Material for statistical analysis performed on movements executed in the same direction relative to gravitational force (S1 Appendix). Focusing on Movement Time, between-subjects analysis revealed that on average each subject tended to move consistently faster or slower than the median across all sets of movements $g^-$, $g^+$, and $g^0$. As the task was self-paced, such result highlights that the duration of movement (MT) is highly subjective and a central behaviour is not derivable across different subjects. However, the same variability is not present for metrics related to movement planning (e.g. PTPV) and smoothness (e.g. SPARC) suggesting for such indicators not to be affected by movement time. Indeed, correlation between MT and PTPV or SPARC did not show any remarkable existing trend. A different conclusion can be drawn for NVP, as slower movements are linked to higher number or peaks in the velocity profile. Please refer to S1 Appendix for detailed presentation of these results.

As Movement Time is a subject-dependent metric, not generalizable across subjects and repetitions, it won't be further discussed in the present manuscript.

Table 3 reports mean±standard deviation (PTPV, PTPSD, and SPARC) or median± inter-quartile range (MT, NVP) for all features but TPE and and MRT for reasons previously indicated.

## Movements across gravity conditions

A qualitative observation of absolute hand paths and speeds for movements against gravity, propelled by gravity, and neutral suggest a similarity between $g^-$ and $g^0$ conditions and a noticeable difference between the two and $g^+$. Therefore the advanced hypothesis is that the selected kinematic features are influenced by the direction of movements with respect to gravitational force (Table 2).

**Percent time to peak velocity.**   PTPV metric across each subject and repetition presented overall a normal distribution, therefore parametric tests were employed (Fig 5A).

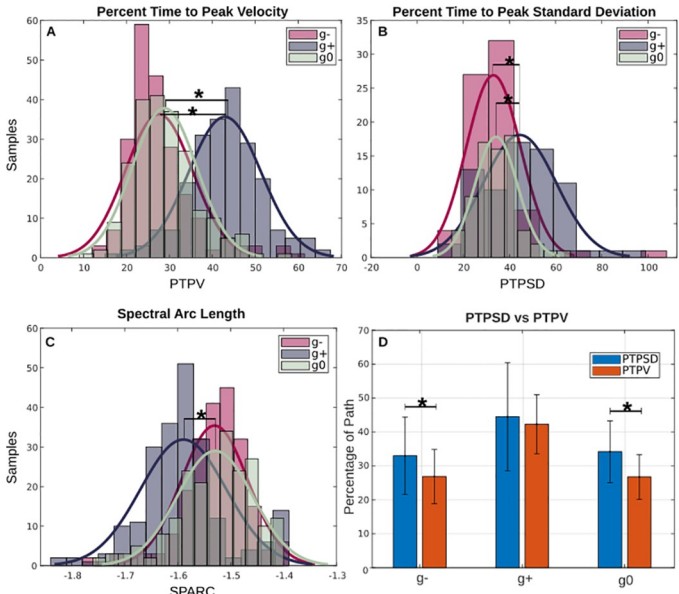

**Fig 5. Statistical distribution of kinematic metrics.** Graphs A, B, and C show the statistical distribution of PTPV, PTPSD, and SPARC metrics respectively. Each metric was calculated for three sets of movements (i.e. g-, in pink, g+, in violet, g0, in green). From these graphs it's possible to notice how the distributions for PTPV, PTPSD, and SPARC metrics are comparable between $g^-$ and $g^0$, as the means are mostly aligned with respect to $g^+$. Graph D reports the comparison between PTPV (in red) and PTPSD (in blue) for each gravity condition. Overall, maximum peak is reached before maximum standard deviation across subjects, trials, and movements.

Across gravity conditions PTPV is comparable between $g^-$ and $g^0$, meanwhile these two are both different from $g^+$.

Repeated measures ANOVA test calculated over the three gravity conditions $g^-$, $g^+$, and $g^0$, did not show a statistically significant effect (p = 0.068, $F(8) = 3.54$, $\eta_p^2 = 0.049$). Pairwise t-tests tests performed over $g^- - g^+$, $g^+ - g^0$ resulted in a statically significant difference (p < 0.0001, d = -1.88 and d = -1.76 respectively), meanwhile $g^- - g^0$ resulted in non-statistical difference (p = 0.078, d = -0.17).

**Percent time to peak standard deviation.** PTPSD metric across each subject and repetition presented overall a normal distribution, therefore parametric tests were employed (Fig 5B).

Across gravity conditions PTPSD is comparable between $g^-$ and $g^0$, meanwhile these two are both different from $g^+$.

Repeated measures ANOVA test calculated over the three gravity conditions $g^-$, $g^+$, and $g^0$, did not show a statistically significant effect (p = 0.97, $F(2) = 0.49$, $\eta_p^2 = 0.0071$). Pairwise t-tests tests performed over $g^- - g^+$, $g^+ - g^0$ resulted in a statically significant difference (p < 0.0001, d = -0.82 and d = -0.79 respectively), meanwhile $g^- - g^0$ resulted in non-statistical difference (p = 0.50, d = -0.11).

Fig 5D reports the comparison for each direction-dependent set of movements between PTPV (in red) and PTPSD (in blue). Overall, for $g^{-,+,0}$ the maximum peak in velocity is reached before the highest variability across repetitions and subjects for each movement. Pairwise t-tests comparisons for PTPV and PTPSD (significance level 0.05) resulted in a statistically significant difference for $g^-$ and $g^0$ (p < 0.001, d($g^-$ = 0.61, d($g^0$ = 0.67)) but not for $g^+$ (p = 0.23, d($g^+$ = 0.14).

**Spectral arc length.** SPARC metric across each subject and repetition presented overall a normal distribution, therefore parametric tests were employed (Fig 5C).

Across gravity conditions SPARC is comparable between $g^-$ and $g^0$, meanwhile these two are both different from $g^+$.

Repeated measures ANOVA test calculated over the three gravity conditions $g^-$, $g^+$, and $g^0$, did not show a statistically significant effect (p = 0.092, $F(2) = 2.43$, $\eta_p^2 = 0.007$). Pairwise t-tests tests performed over $g^- - g^+$, $g^+ - g^0$ resulted in a statically significant difference (p < 0.0001, d = 0.81 and d = 0.78 respectively), meanwhile $g^- - g^0$ resulted in non-statistically significant difference (p = 0.88, d = -0.014).

**Target position error and minimum required tunnel.** TPE and MRT metrics were not subjected to any statistical analysis for the reasons previously reported. Considering TPE, conditions $g^-$ and $g^+$ exhibit comparable values ($g^-$ of 3.33 cm, $g^+$ 3.3 cm), meanwhile for $g^0$ the TPE is 2.3 cm (Fig 6A). Instead, for MRT $g^-$ exhibited the lower values (3.84 cm) with respect to $g^+$ and $g^0$, 4.25 cm and 4.69 cm respectively (Table 3). This is in line with Fig 6B of the trends of 1 standard deviation (solid curve) and 3 standard deviations (dashed curve) across subjects, repetitions and movements. A qualitatively observation of the graphs highlights how $g^0$ presents a higher standard deviation than $g^{-,+}$ thus a higher integral between the two curves (Table 3).

**General conclusion.** In summary, the results suggest that the hypothesis regarding the impact of movement direction in relation to gravity on the selected metrics is not disproved when comparing $g^-$ to $g^+$ for PTPV, SPARC, PTPSD, NVP, MRT and it is rejected for TPE. Similar conclusions can be drawn for $g^0$ to $g^+$ but, different from previous observations, TPE and MRT both show a direction-dependent outcome. Finally, the hypothesis is rejected for $g^-$ to $g^0$ comparison for all the metrics (i.e. PTPV, SPARC, PTPSD, NVP) but TPE and MRT.

## Discussion

Neurological disorders can result in significant impairments of the upper-limb, thereby restricting daily activities. Essential tasks such as organizing, eating, and maintaining personal hygiene often transpire within the Sagittal plane, necessitating consistent movements of the shoulder and elbow, and interaction with loads. The influence of gravitational force is pivotal in the planning and execution of these movements. In this study, we propose a user-friendly setup to examine healthy postural patterns during the execution of functional tasks. Specifically, the pick-and-place movement serves as a simplified representation of everyday activities, offering an avenue to assess user performance in task completion.

In the analysis of MoCap data acquired from healthy subjects, the investigation of healthy postural patterns during the pick-and-place movement revealed recurrent features. Results concerning absolute hand positions indicated a minimal Root Mean Square Error—on the order of millimeters—between the experimental curve and a 6th order polynomial fitting function. This aligns with expectations for smooth movements, as supported by existing literature on motor coordination [42, 71]. Notably, considering the measurements in the experimental setup span approximately 40 cm, the observed error can be considered negligible (Fig 3A).

The profile of absolute hand positions in $g^-$ closely resembles that in $g^0$, contrasting with the pattern observed in $g^+$. The main glaring difference can be found in the corrective stage, less remarked in movements propelled by gravitational force. Indeed, $g^+$ solely showed the start of the correction phase at 87% of the path, meanwhile $g^-$ and $g^0$ at 60%. Such result is in contrast with the study of Lyons et al. [72] where for rapid aiming movements $g^+$ would present a higher corrective stage.

A similar, more subtle difference can be also observed during the movement phase. This indicates that in the proposed task a different model for motor planning and execution is created when facing unrestrained upward and downward movement. These observations align

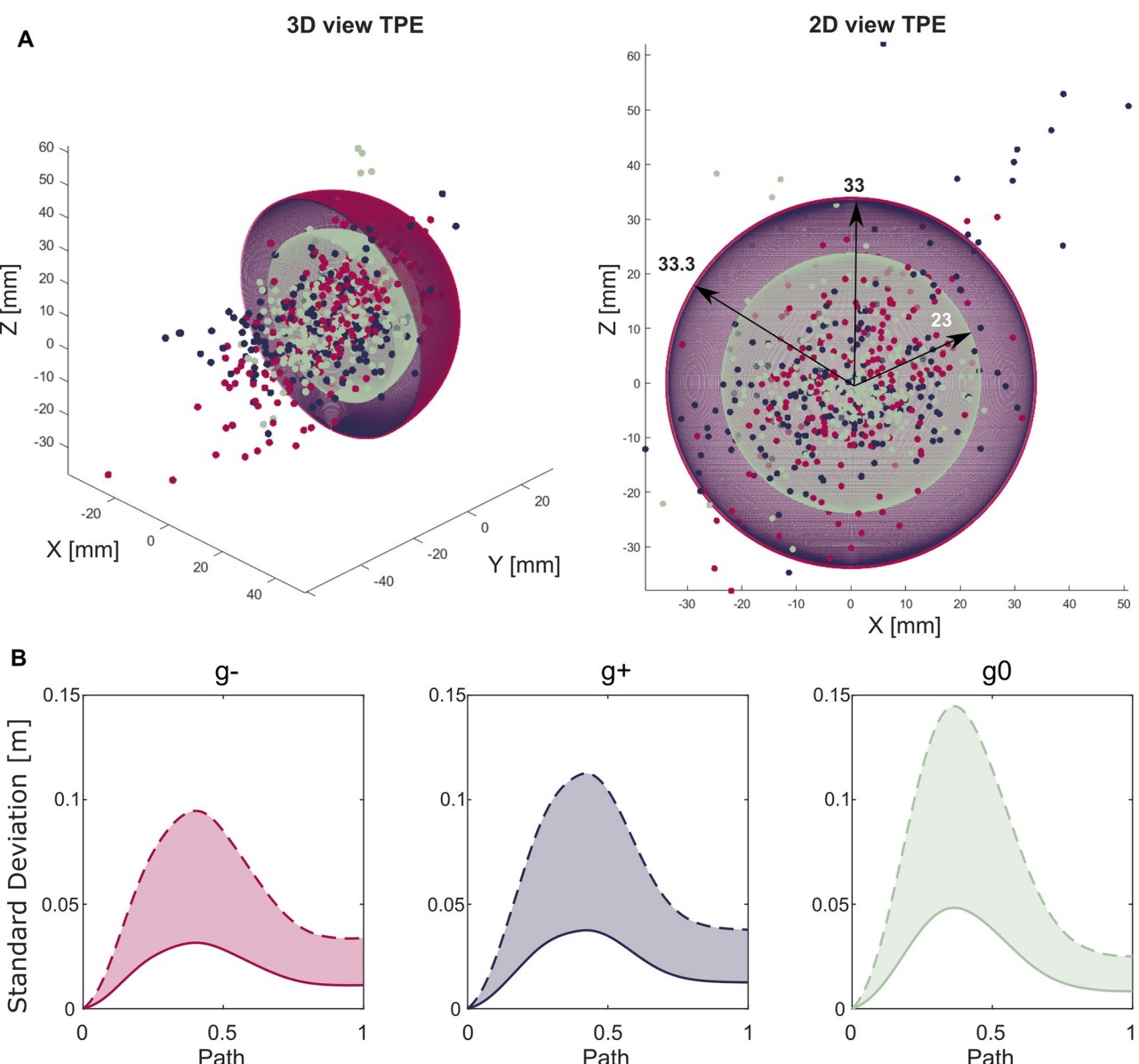

**Fig 6. Representation of Target Position Error and Minimum Required Tunnel metrics.** (A) 3D and 2D view of Target Position Error (TPE) metric defined as the sphere whose radius contains up to 95% of hand end-points. Sphere related to $g^-$ (in pink) and $g^+$ (in violet) are almost symmetrical and present radius of 3.33 and 3cm respectively. $g^0$ sphere (in green) radius is 2.3cm. The small dots represent experimental data, each colored in accordance to the gravity condition they belong to. (B) Representation of the average standard deviations across subjects, trials and movements for $g^-$, $g^+$, and $g^0$. Solid line is representative of 1 standard deviation, meanwhile dashed line of 3 standard deviations. The Minimum Required Tunnel (MRT) metric is defined as the integral between solid and dashed lines (Table 3), shaded in the graph.

with the conclusions drawn by Atkeson and colleagues [10] regarding the computation of reaching arm trajectories between distinct points in the vertical plane. Focusing on each movement, it is evident that the trend of each curve is comparable but not exactly superimposed across subjects and repetitions (Fig 3). However, the percent time at which peak standard deviation is reached (defined as PTPSD) does not depend on the start and end position of the object in the same gravity condition ($p > 0.017$), but rather presents statistical difference when comparing upward ($g^-$) and downward movements ($g^+$) as $p < 0.0001$, visible in Fig 5 (Table 4).

**Table 4. Statistical analysis on the distribution of each set of movement ($g^-$, $g^+$, $g^0$).** Repeated measures ANOVA analysis was performed for NVP, PTPV, SPARC, and PTPSD metrics. Friedman test was used for NVP ($\alpha$ = 0.05). Pair-wise Post-hoc t-student test were conducted for PTPV, PTPSD, and SPARC metric, post-hoc Mann-Whitney test for MT (Bonferroni correction, $\alpha$ = 0.017).

| Metrics | Global gravity effect | Pair-wise post-hoc tests | | |
|---|---|---|---|---|
| | | $g^- - g^+$ | $g^- - g^0$ | $g^+ - g^0$ |
| PTPV | 0.067 | <0.0001* | 0.078 | <0.0001* |
| NVP | <0.0001* | 0.006* | 0.86 | 0.011* |
| SPARC | 0.092 | <0.0001* | 0.88 | <0.0001* |
| PTPSD | 0.97 | <0.0001* | 0.5 | <0.0001* |

*Statistical significance.

Similar conclusions can be drawn for absolute velocity profiles (Fig 3), where the outline of $g^-$ and $g^0$ is comparable and different from $g^+$. Focusing the attention on the percent time when peak velocity is reached (referred as PTPV), $g^{-,0,+}$ do not present statistically significant comparable location of maximum velocity in each gravity condition, but they do when comparing upward to downward movements as p < 0.0001 (Table 4. For a visual representation please refer to Fig 4A) and downward to neutral movements (p < 0.0001). Meanwhile comparing $g^-$ to $g^0$ conditions did not yield any statistical difference (p = 0.078). This is line with the state-of-the-art related to changes in direction-dependent movements [73]—Indeed, Papaxanthis et al. [5] remarked that the velocity trend of fingertip movements did not remain constant in vertical motions and attributed this variation to the influence of gravity on the programming processes of the central nervous system.

The shape of the absolute hand velocity holds meaningful information regarding the *smoothness* of the performed movement and can be exploited as a marker for neurological disorders evaluation. Point-to-point reaching tasks in healthy subjects result in bell shaped curves presenting only one peak as the movement is characterized by only one acceleration and deceleration pattern, namely submovement. The same task performed by post-stroke survivors led to an abnormal speed profile characterized by several peaks [33, 34], thus several submovements, converging into the healthy behaviour following a rehabilitative journey. This suggests for the number of peaks in the velocity profile (referred to as NVP) to be a smoothness indicator. However, the proposed study requires dynamic actions in the 3D space while interacting with a load. Nonetheless, a quantification of Number of Velocity Peaks is needed to provide insights on the motor-control strategies in performing the task, as well as to exploit it as a kinematic metric of upper-limb impairments. For all gravity conditions the majority of speed outlines presented only one peak, coincident with the maximum velocity (Fig 4). However, $g^+$ presents a comparable number of trajectories with one or two peaks, indicating that movements performed propelled by gravity in the frontal plane are less smooth than upward ones. Indeed, statistical analysis confirmed for NVP to not be remarkably different between $g^{-,0}$ as p = 0.86, but significant for $g^{+,0}$ and $g^{-,+}$ as p = 0.011 and p = 0.006 respectively.

This is reflected in the Spectral Arc Length metric (referred as SPARC), where for $g^+$ is more negative and statistically different than $g^{-,0}$ as p < 0.0001 (Tables 3 and 4, Fig 5). A similar trend was observed in the study of Bayle et al. [56] for reaching movements in the 3D space, where backward movements presented lower SPARC than forward ones. Such behaviour may be attributed to different muscle synergies employed during the two movements. In point-to-point reaching tasks, the coordinated activation of agonist and antagonist muscles is needed for creating a bell-shaped speed profile. Without this precise timing, the resultant movement lacks smoothness [74, 75]. Current results seem to suggest that downward

movements exhibit a lower muscle coordination than upward ones. Comparison of SPARC data in the present study and data found in literature [56, 63] indicate that, overall, the proposed pick-and-place task lead to less smooth profiles than free reaching (either 3D or 2D). This might be due to the difference in nature of the task and the additional interaction with physical objects.

The *efficiency* of the movement has been extensively linked in literature to time spent to perform it (referred as Movement Time). Movements performed in the three conditions presented a duration higher than 1 second and do not follow a normal distribution. This was expected since subjects were not given any restriction on this matter. The positive skew in the trends reveal that subjects tended to perform the task rather slower than the average. However, a between-subjects analysis underlined a strong subjectivity in task duration for $g^{-,+,0}$ and a difficulty in characterizing a central tendency in the acquired dataset. Therefore, analyses on MT were not further computed as the aim of the study was to define the role of gravity on normative markers for the proposed pick-and-place set-up. Moreover, MT does not affect the other metrics but NVP as not specific trends exist between them. Please refer to S1 Appendix for further comments.

Moving onto metrics related to movement *planning*, both Percent Time to Peak Velocity and Percent Time to Peak Standard Deviation present statistical difference between $g^{-, 0}$ and $g^{+}$ (p < 0.0001) but not between $g^-$ and $g^0$ as p = 0.078 and p = 0.5 respectively. Fig 5 shows the comparison between percentage time at which maximum velocity is reached with respect to maximum standard deviations across subjects and repetitions. As it's possible to notice, PTPV (in red) is reached before PTPSD (in blue) in all conditions. A possible interpretation could be the trade-off between speed and accuracy. In accordance with Fitt's Law, as the speed of a movement increases, accuracy relative to a reference trajectory decreases [76]. The idea would be that all subjects begin the movement in a rather comparable way. Then, when accelerating toward the target, the speed increases (with a peak at Percent Time to Peak Velocity) resulting in an increase of deviations of hand paths across each curve (with a peak at Percent Time to Peak Standard Deviation), thus a higher standard deviation. Ideally it would be expected for the PTPV to coincide with PTPSD as maximum speed should lead to maximum deviations. This is true for $g^+$ as statistical analysis conducted between the two distributions yielded p = 0.23 (significance level of 0.05), but not for $g^-$ and $g^0$ which presented p < 0.001.

Converting the difference between Percent Time to Peak Velocity and Percent Time to Peak Standard Deviation from fraction of path to actual time duration in second, the registered latency was on average 86 ms. During the planning of a motor task, visual inputs are transformed into motor command [77]. Particularly, the human eye is usually planned and launched between 150 to 200 ms [78], thus the registered latency of 86 ms can be considered negligible. Percent Time to Peak Velocity and Percent Time to Peak Standard Deviation behaviour is in line with Fitt's law [76, 79].

Focusing on the *accuracy*, two novel indicators were introduced. Particularly, Target Position Error refers to the the accuracy in placing the load on target, meanwhile Minimum Required Tunnel relates to the accuracy throughout movement execution across subjects and repetitions. Target Position Error, did not present the same trend of metrics previously described, but rather a similarity between movements propelled and against gravity ($g^{-,+}$) with respect to neutral $g^0$. Again, such result differs from studies in literature [72, 80] as movements propelled by gravity were characterized by higher energy consumption and related higher endpoint error. Indeed, Target Position Error for objects picked from the table and placed on a second target on the table is 10 mm smaller than the other two conditions (Fig 6A).

This might be attributed to the underlying principles of hand-eye coordination. The receptors in the retina provide useful feedback on the position of the target with respect to the

position of the limb. As a consequence, the plans adopted by the central nervous system to correct the reaching movement highly depend on visibility of both the hand and the target at the same time [81] and on intermittent motion impulses related to the visual and proprioceptive feedback [82]. This is due to a theory that the central nervous system tends to "forget" the position of the target if not constantly visible. Considering the experimental set-up (Fig 1), $g^0$ does not require large vertical head movements to visualize both the start and target position of the object, as well as the hand while moving it. This is not true for movements $g^{-,+}$ thus explaining why TPE is higher in those two conditions.

Concerning MRT, Table 2 reports that $g^-$ presents a lower MRT than $g^+$, suggesting that large vertical movements performed against gravity and away from the body lead to less intra-subject variability than movements performed propelled by gravity and toward the body ($g^+$). Focusing on the performance of the task with respect to the body, the study of Hajihosseinali and colleagues [57] on a 2D point-to-point horizontal task reported that movements away from the body were more repeatable than movements toward the body. A possible reason can be found in the neurophysiology of the human brain. The primary motor cortex is in charge of the planning and execution of voluntary movements. Motor cortical cells in the primary motor cortex of primates are characterized by a preferred direction at which their discharge rate peaks.

The study of Naselaris et al. [83] for 3D, unconstrained, reaching movements highlighted that away-from-the-body movements elicited an higher level of activation of the primary motor cortex than toward-the-body, thus suggesting a less coordinated movement for the latter, also resulting in a less smooth path. As a consequence, Hajihosseinali et al. [57] encourages the assessment of neurological patients through away-from-the-body tasks, more challenging and representative of the patient's movement quality. Since the presented task requires for $g^-$ to move the objects from a position close to the body and to place them forward on a shelf, and for $g^+$ the exact opposite, results related to the MRT seem in line with the state-of-the-art of direction-dependent cortical activation.

On the contrary, $g^0$ reports the highest MRT with respect to $g^{-,+}$. This result was expected as the task was performed differently between subjects. Indeed, some participants tended to bring the picked object closer to their body performing a movement in the Transversal plane, meanwhile others lifted it on the vertical axis and preferred a movement in the Sagittal plane. Interestingly, such difference in performance did not affect *planning* and *smoothness* metrics.

While Target Position Error and Minimum Required Tunnel were calculated based on all trajectories per type of movement, resulting in only three evaluations for each gravity condition, it's important to note that these metrics are not intended to serve as definitive measures of central tendency for movement accuracy among healthy subjects. Rather, they offer an indicator of task-specific global accuracy within this population. In the future, when compared to data from non-healthy subjects, these metrics may provide direct insights into differences in task performance.

## Conclusions

Starting from a novel and simple experimental set-up for upper-limb functional recovery assessment, the present study aimed at characterizing the kinematic strategies applied by a population of healthy subjects when performing a pick-and-place. Different from free-reaching tasks extensively proposed in literature, the interaction with a load while performing a movement can lead to positive effects on functional performance of neurologically impaired subjects [84]. A similar, interesting, study was carried about by Valevicius et al. [64], where two novel standardized activities of daily living tasks for the upper-limb characterization of

hand movements are presented. The authors focused on both reach-to-grasp and transport-release phases of loads and calculated several kinematic metrics based on hand paths, velocities, and grip aperture. While authors provide some comments on how the selected features change according to movements performed, little contextualization was provided with respect to the state-of-the-art of motor control strategies. Moreover, no specific attention was paid on the effect of gravity in performing the movements, which greatly influences the proposed tasks. On the contrary, projecting the focus of the studies of upper-limb motor behavior onto the principals of motor-control can be crucial to discuss applicability of research findings, inform rehabilitation strategies, and contribute to the ongoing advancement of knowledge in the field.

The limitations of the present study include the recruitment of relatively young participants in the study and the proposed metrics should be exploited as reference for age-matched patients population. As neurological impairments (i.e., stroke) mostly involve senior population, future studies should involve the participation of elder, able-bodied, individuals to create a secondary, suitable, kinematic dataset. Any change in kinematic behaviour of elderly subjects' hand could be investigated, as previous studies highlighted changes in vertical movements in elderly population [85–87].

Moreover, the acquired data constitute of a balanced sex distribution, but forthcoming investigations could focus on the inquiry of any, if present, motor-control strategy differences between female and male participants.

Additional studies will be performed on inter-joint coordination of the upper-limb (i.e. Shoulder, Elbow, and Wrist displacements) while executing the proposed pick-and-place task, with the aim to identify useful metrics for the discrimination of compensation from actual recovery.

Observations pertaining to healthy hand displacements in the execution of a pick-and-place task have brought to light the substantial influence exerted by diverse gravity conditions on task performance. The execution of movements against or in concert with gravity instigates different motor plans by the central nervous system, giving rise to discernible distinctions in biomechanical behaviors. The outcomes of this study underline the statistical significance in kinematic metrics when concentrating on two gravity conditions among subjects. Consequently, when designing rehab sessions for upper-limb impairments, it's important to keep in considerations the biomechanical differences in these two conditions during evaluations.

Overall, the proposed study presented the characterization of a standardized upper-limb functional movement while interacting with a load. The acquired absolute hand positions and velocities showed distinct behavior for movements performed against or propelled by gravity, in line with previous motor-control studies. From this dataset, five already-in-use and two novel kinematic features were tested and compared with the state-of-the-art to assess the impact of gravity in upper-limb healthy functional movements for metrics useful in rehabilitation.

## Supporting information

**S1 Appendix. Supplementary analyses on gravity-related hand kinematic features.** In this section, additional analyses were conducted related to the impact of gravity on movements performed in the same direction with respect to the gravitational field. Moreover, a within-analyses investigation highlighted, if present, any non-normative behaviour for each subject involved, as well as the correlation between hand spatio-temporal features.
(PDF)

## Acknowledgments

The authors thank Inna Forsiuk for helping in Motion Capture Data collection and processing. The authors would like to thank also Giulia Mariani, Gianluca Capitta, and Paolo Rossi for helping out in building the set-up and providing assistance during the experiments.

## Author Contributions

**Conceptualization:** Anna Bucchieri.

**Data curation:** Anna Bucchieri.

**Formal analysis:** Anna Bucchieri.

**Funding acquisition:** Matteo Laffranchi, Lorenzo De Michieli.

**Investigation:** Anna Bucchieri.

**Methodology:** Anna Bucchieri, Federico Tessari.

**Project administration:** Matteo Laffranchi, Lorenzo De Michieli.

**Supervision:** Federico Tessari.

**Visualization:** Anna Bucchieri.

**Writing – original draft:** Anna Bucchieri.

**Writing – review & editing:** Federico Tessari, Stefano Buccelli, Elena De Momi, Matteo Laffranchi, Lorenzo De Michieli.

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
