## [Decision Letter · Decision Letter 0]

9 Jun 2024

PONE-D-24-10631The effect of gravity on hand spatio-temporal kinematic features during functional movementsPLOS ONE

Dear Dr. Bucchieri,

Thank you for submitting your manuscript to PLOS ONE. After careful consideration, we feel that it has merit but does not fully meet PLOS ONE’s publication criteria as it currently stands. Therefore, we invite you to submit a revised version of the manuscript that addresses the points raised during the review process.

We look forward to receiving your revised manuscript.

Kind regards,

Monika Błaszczyszyn

Academic Editor

PLOS ONE

“This work was supported by Istituto Nazionale per l’Assicurazione contro gli Infortuni sul Lavoro, under grant agreement ”PR19-RR-P2-RoboGYM”.

“The authors thank Inna Forsiuk for helping in Motion Capture Data collection and processing. The authors would like to thank also Giulia Mariani, Gianluca Capitta, and Paolo Rossi for helping out in building the set-up and providing assistance during the experiments. This work was supported by Istituto Nazionale per l’Assicurazione controgli Infortuni sul Lavoro, under grant agreement ”PR19-RR-P2-RoboGYM.”

“This work was supported by Istituto Nazionale per l’Assicurazione contro gli Infortuni sul Lavoro, under grant agreement ”PR19-RR-P2-RoboGYM”.

Reviewers' comments:

Reviewer's Responses to Questions

**Comments to the Author**

1. Is the manuscript technically sound, and do the data support the conclusions?

Reviewer #1: Yes

Reviewer #2: Yes

Reviewer #3: Yes

2. Has the statistical analysis been performed appropriately and rigorously? 

Reviewer #1: Yes

Reviewer #2: Yes

Reviewer #3: Yes

3. Have the authors made all data underlying the findings in their manuscript fully available?

Reviewer #1: No

Reviewer #2: Yes

Reviewer #3: Yes

4. Is the manuscript presented in an intelligible fashion and written in standard English?

Reviewer #1: Yes

Reviewer #2: Yes

Reviewer #3: Yes

5. Review Comments to the Author

Reviewer #1: he manuscript submitted for consideration seeks to assess the the effect of gravity on hand spatio-temporal kinematic features during functional movements. It is evident that the manuscript has been meticulously crafted and organized. The study exhibits a sound methodology and effectively addresses the research questions it posits. In my opinion, this article displays strong potential for publication in a reputable journal.

Reviewer #2: The paper is well written and sound, I'm favourable to its publication after addressing a few comments:

• Line 276. Here, the following papers in which the authors analysed upper limb movement in Parkinson’s disease may be useful for you:

o Cesarelli, G., Donisi, L., Amato, F., Romano, M., Cesarelli, M., D’Addio, G., ... & Ricciardi, C. (2023). Using features extracted from upper limb reaching tasks to detect Parkinson’s disease by means of machine learning models. IEEE Transactions on Neural Systems and Rehabilitation Engineering, 31, 1056-1063.

o Ponsiglione, A. M., Ricciardi, C., Amato, F., Cesarelli, M., Cesarelli, G., & D’Addio, G. (2022). Statistical analysis and kinematic assessment of upper limb reaching task in Parkinson’s disease. Sensors, 22(5), 1708.

•Line 498. It would be of interest having some information regarding the MoCap protocol proposed by Vicon and employed for acquiring the subjects.

• Line 567. “…this metrics was…” should be “this metrics was”.

• Line 629. “Statistical analysis were performed…” would better be “Statistical analysis was performed”.

• In the "Statistical Analysis" section, the authors explain they use different test according to the normality of the data. How did they check the normality distribution of the data?

Reviewer #3: The article investigates variations in upper limb movements in relation to their direction with respect to gravity. Experiments with 24 healthy subjects were conducted. In these, the participants performed object movement tasks between different heights on a shelf. Traditional metrics were used in the analyses, and two new metrics were proposed.

The extent and quality of the analysis of the results and discussions, with comparisons to related works, are highlighted, indicating new findings, and results consistent with those reported in correlated works, and pointing out hypotheses for those that are discordant. Discussions and conclusions about the new metrics are also presented (more comments concerning the conclusions are further provided). Thus, the article presents a significant scientific contribution to the field of upper limb biomechanics and its related areas.

The text has both major and minor deficiencies. Overall, it is well written; however, a revision of the English, and improvements in the writing and organization of the text, are necessary.

The following major improvements are suggested:

- Reorganization of the text. Normally, articles involving experiments of this nature follow the order: Introduction, Materials and Methods, Results and Analyses, Discussions, and Conclusion. The current text presents a different order, which is considered unusual by this reviewer;

- Inclusion of a conclusions section. It is considered that the text from line 437 to 484 constitutes a good text for the conclusions;

- References have to be numbered according to the order in which they appear in the text;

- Why is the only cited robotic rehabilitation system the MIT-Manus? Is there any reason for this preference? There are many other robotic rehabilitation systems. There are excellent review articles on robotic systems for rehabilitation that could be cited instead;

- The content of the abstract is faithful to the text; however, important information can be added, such as more details about the experiments and the main results and conclusions of the study;

- Paragraph from line 137 up to 143. Axis of a reference system is mentioned. How is the reference system oriented in the experimental environment?

Minor corrections:

- Is the "Author Summary" section necessary? In this reviewer's opinion, this writing style is unusual for papers;

- The text uses an excess of abbreviations and acronyms. Remove those that appear infrequently, such as ARAT, FMA, RMSE, etc. Keep only those that appear frequently and throughout the text;

- The repetition of acronyms and abbreviations very close to each other in a text should be avoided, just as the repetition of words is. Reducing acronyms and abbreviations improves the fluency of reading. Review the text, especially the results section

- In Figure 1, markers are named with the same format as the references citation, which may lead to misunderstanding;

- Some paragraphs are too long, such as those from line 2 to 61 and from line 386 to 429. These paragraphs contain more than one subject. Splitting them according to more correlated and few subjects may improve writing quality.

Considering the scientific contribution of the results, novel metrics, and discussion, it is recommended the paper be accepted after the suggested improvements.

6. PLOS authors have the option to publish the peer review history of their article (what does this mean?). If published, this will include your full peer review and any attached files.

Reviewer #1: No

Reviewer #2: No

Reviewer #3: **Yes: **Leonardo Marquez Pedro

---

## [Author Response · Author response to Decision Letter 0]

11 Aug 2024

Dear Dr. Monika Błaszczyszyn,

We would like to thank the reviewers and the editor for their constructive feedback on our manuscript entitled "The effect of gravity on hand spatio-temporal kinematic features during functional movements". We believe the revised manuscript addresses all the comments, questions and concerns of the reviewers, and represents a significant improvement to the previously submitted manuscript. We invite the editors and reviewers to inspect the revised manuscript as well as the response to the reviewers for a detailed explanation of all the papers' modifications.

According to the decision letter, the following statements are to be clarified:

1. The funders had no role, thereby we would like to state “The funders had no role in study design, data collection and analysis, decision to publish, or preparation of the manuscript.”

2. Funders information were removed from the Acknowledgments Section of the manuscript.

3. The dataset acquired and analyzed in the present manuscript is currently under review from the Research Data Management department of the Italian Institute of Technology (https://dataverse.iit.it). Once approved, the dataset will be public and a DOI will be generated.

Reviewer 1:

The manuscript submitted for consideration seeks to assess the the effect of gravity on hand spatio-temporal kinematic features during functional movements.

It is evident that the manuscript has been meticulously crafted and organized. The study exhibits a sound methodology and effectively addresses the research questions it posits. 

In my opinion, this article displays strong potential for publication in a reputable journal.

reply:

Thank you for your positive and encouraging feedback on our manuscript. We are delighted that you found the organization, methodology, and overall approach of our study to be of high quality. 

We appreciate your recognition of the potential impact of our work and your support for its publication in a reputable journal. Should you have any further suggestions or specific areas where you believe additional enhancements could be made, we would be grateful for your input.

Reviewer 2:

The paper is well written and sound, I'm favorable to its publication after addressing a few comments:

Line 276. Here, the following papers in which the authors analyzed upper limb movement in Parkinson’s disease may be useful for you:

o Cesarelli, G., Donisi, L., Amato, F., Romano, M., Cesarelli, M., D’Addio, G., ... & Ricciardi, C. (2023). Using features extracted from upper limb reaching tasks to detect Parkinson’s disease by means of machine learning models. IEEE Transactions on Neural Systems and Rehabilitation Engineering, 31, 1056-1063.

o Ponsiglione, A. M., Ricciardi, C., Amato, F., Cesarelli, M., Cesarelli, G., & D’Addio, G. (2022). Statistical analysis and kinematic assessment of upper limb reaching task in Parkinson’s disease. Sensors, 22(5), 1708.

reply:

We thank the reviewer for suggesting additional literature in support of upper limb functional assessment in people affected by a neurological disorder. The proposed papers were added in the Introduction section when addressing the topic of upper-limb recovery in people affected by an impairment.

•Line 498. It would be of interest having some information regarding the MoCap protocol proposed by Vicon and employed for acquiring the subjects.

We thank the reviewer for suggesting the need to add further information on the adopted MoCap protocol. Details were provided in the Materials and Methods section as follows:

reply:

MoCap cameras were calibrated following Nexus Vicon guidelines. The calibration was performed for each subject to ensure accuracy in kinematic data acquisition of the reflective markers. The global reference frame was set on the floor in correspondence to the bottom left corner of the custom-made library (Fig 2).

• In the "Statistical Analysis" section, the authors explain they use different test according to the normality of the data. How did they check the normality distribution of the data?

reply:

We thank the reviewer for pointing out the missing information about the normality test exploited in this study. The Statistical Analysis section was updated with the following information: 

Normality of the dataset was assessed employing the Jarque-Bera test.

Reviewer 3:

The following major improvements are suggested:

- Reorganization of the text. Normally, articles involving experiments of this nature follow the order: Introduction, Materials and Methods, Results and Analyses, Discussions, and Conclusion. The current text presents a different order, which is considered unusual by this reviewer;

- Inclusion of a conclusions section. It is considered that the text from line 437 to 484 constitutes a good text for the conclusions;

- References have to be numbered according to the order in which they appear in the text;

- Is the "Author Summary" section necessary? In this reviewer's opinion, this writing style is unusual for papers;

reply:

We thank the reviewer for pointing out the need of reorganizing the text. We apologize for looking over the sections order proposed by PLOS ONE.

The sections were reorganized following PLOS ONE Authors guidelines. Moreover, a Conclusion section was added. References were numbered correctly according to the order of citation. Author Summary was removed from the manuscript.

- Why is the only cited robotic rehabilitation system the MIT-Manus? Is there any reason for this preference? There are many other robotic rehabilitation systems. There are excellent review articles on robotic systems for rehabilitation that could be cited instead;

reply:

We thank the reviewer for their comment related to the MIT-Manus. Such system was cited in the context of recreating standard and reproducible kinematic behaviour when performing planar reaching tasks. The same level of standardization is not possible when considering multi-DOF robotic systems allowing movements in the 3D space. Additional systems were cited in the manuscript as suggested by the reviewer.

- The content of the abstract is faithful to the text; however, important information can be added, such as more details about the experiments and the main results and conclusions of the study;

reply:

We thank the reviewer for highlighting the lacking of important information in the abstract. The latter was updated in the manuscript as follows:

Understanding the impact of gravity on daily upper-limb movements is crucial for comprehending upper-limb impairments. This study investigates the relationship between gravitational force and upper-limb mobility by analyzing hand trajectories from 24 healthy subjects performing nine pick-and-place tasks, captured using a motion capture system. The results reveal significant differences in motor behavior in terms of planning, smoothness, efficiency, and accuracy when movements are performed against or with gravity. Analysis showed that upward movements (g-) resembled transversal ones (g0) but differed significantly from downward movements (g+). Corrective movements in g+ began later than in g- and g0, indicating different motor planning models. Velocity profiles highlighted smoother movements in g- and g0 compared to g+. Smoothness was lower in g+, indicating less coordinated movements. Efficiency showed significant variability with no specific trends due to subjective task duration among subjects.

\\This study highlights the importance of considering gravitational effects when evaluating upper-limb movements, especially for individuals with neurological impairments. Planning metrics, including Percent Time to Peak Velocity and Percent Time to Peak Standard Deviation, showed significant differences between g- and g0 compared to g+, supporting Fitts' law on the trade-off between speed and accuracy. Accuracy indicators, such as Target Position Error and Minimum Required Tunnel novelly introduced in this work, provided insights into hand-eye coordination and movement variability. The findings suggest that motor planning, smoothness, and efficiency are significantly influenced by gravity, emphasizing the need for differentiated approaches in assessing and rehabilitating upper-limb impairments. Future research should explore these metrics in impaired populations to develop targeted rehabilitation strategies.

- Paragraph from line 137 up to 143. Axis of a reference system is mentioned. How is the reference system oriented in the experimental environment?

We thank the reviewer for suggesting the need to add information about the reference system. Additional information were provided in the manuscript as follow:

reply:

MoCap cameras were calibrated following Nexus Vicon guidelines. The calibration was performed for each subject to ensure accuracy in kinematic data acquisition of the reflective markers. The global reference frame was set on the floor in correspondence to the bottom left corner of the custom-made library (Fig 2)

Moreover, Fig 1 and Fig 2 were updated with a visual representation of reference systems. 

- In Figure 1, markers are named with the same format as the references citation, which may lead to misunderstanding;

reply:

We thank the reviewer for pointing out the possible misunderstanding in Fig 1. Labels were changed to avoid any confusion with the reference citation.

- The text uses an excess of abbreviations and acronyms. Remove those that appear infrequently, such as ARAT, FMA, RMSE, etc. Keep only those that appear frequently and throughout the text;

- The repetition of acronyms and abbreviations very close to each other in a text should be avoided, just as the repetition of words is. Reducing acronyms and abbreviations improves the fluency of reading. Review the text, especially the results section

- Some paragraphs are too long, such as those from line 2 to 61 and from line 386 to 429. These paragraphs contain more than one subject. Splitting them according to more correlated and few subjects may improve writing quality.

reply:

We thank the review for the remark. It was decided to keep only abbreviations related to kinematic metrics in the manuscript and not to report acronyms related to other quantities or tests. The paragraphs were also split as suggested by the reviewer to increase the manuscript readability.

---

## [Decision Letter · Decision Letter 1]

27 Aug 2024

The effect of gravity on hand spatio-temporal kinematic features during functional movements

PONE-D-24-10631R1

Dear Dr. Bucchieri,

We’re pleased to inform you that your manuscript has been judged scientifically suitable for publication and will be formally accepted for publication once it meets all outstanding technical requirements.

Kind regards,

Monika Błaszczyszyn

Academic Editor

PLOS ONE

Additional Editor Comments (optional):

Reviewers' comments:

Reviewer's Responses to Questions

**Comments to the Author**

1. If the authors have adequately addressed your comments raised in a previous round of review and you feel that this manuscript is now acceptable for publication, you may indicate that here to bypass the “Comments to the Author” section, enter your conflict of interest statement in the “Confidential to Editor” section, and submit your "Accept" recommendation.

Reviewer #3: All comments have been addressed

2. Is the manuscript technically sound, and do the data support the conclusions?

Reviewer #3: Yes

3. Has the statistical analysis been performed appropriately and rigorously? 

Reviewer #3: Yes

4. Have the authors made all data underlying the findings in their manuscript fully available?

Reviewer #3: Yes

5. Is the manuscript presented in an intelligible fashion and written in standard English?

Reviewer #3: Yes

6. Review Comments to the Author

Reviewer #3: All the suggestions to clarify research details and to improve the text were satisfactorily implemented, or proper justifications were given by the authors.

7. PLOS authors have the option to publish the peer review history of their article (what does this mean?). If published, this will include your full peer review and any attached files.

Reviewer #3: **Yes: **Leonardo Marquez Pedro

---

## [Editor Report · Acceptance letter]

11 Oct 2024

PONE-D-24-10631R1 

PLOS ONE

Dear Dr. Bucchieri, 

I'm pleased to inform you that your manuscript has been deemed suitable for publication in PLOS ONE. Congratulations! Your manuscript is now being handed over to our production team.

Kind regards, 

on behalf of

Dr. Monika Błaszczyszyn 

Academic Editor

PLOS ONE